# Prognostic Value of Immunohistochemical T-Cell Marker Loss in Early-Stage Mycosis Fungoides: A Single-Center Cohort Study

**DOI:** 10.3390/dermatopathology12030029

**Published:** 2025-09-10

**Authors:** Sandra Jerkovic Gulin, Ivana Ilic, Dario Gulin, Georgios Kravvas, Romana Ceovic

**Affiliations:** 1Department of Dermatology and Venereology, Ryhov County Hospital, 553 05 Jonkoping, Sweden; 2Department of Biomedical and Clinical Sciences, Faculty of Medicine and Health Sciences, Linkoping University, 581 83 Linkoping, Sweden; 3Department of Pathology and Cytology, University Hospital Centre Zagreb, 10 000 Zagreb, Croatia; ricilic@gmail.com; 4Department of Internal Medicine, Ryhov County Hospital, 553 05 Jonkoping, Sweden; 5Department of Health, Medicine and Caring Sciences, Linköping University, 581 83 Linkoping, Sweden; 6Department of Dermatology, University College London Hospitals NHS Foundation Trust, London NW1 2BU, UK; georgios.kravvas@nhs.net; 7Department of Medicine, University College London, London NW1 2BU, UK; 8Department of Dermatology and Venereology, School of Medicine, University Hospital Centre Zagreb, University of Zagreb, 10 000 Zagreb, Croatia

**Keywords:** mycosis fungoides, CD7, immunohistochemistry, T-cell markers, prognosis, progression-free survival, early-stage cutaneous lymphoma, CD2, CD5, survival analysis

## Abstract

Introduction: Mycosis fungoides (MF) is the most common cutaneous T-cell lymphoma, often exhibiting loss of pan-T-cell markers such as CD2, CD3, CD5, and CD7. While these immunophenotypic alterations assist in diagnosis, their prognostic relevance in early-stage MF remains uncertain. This study aimed to determine whether immunohistochemical loss of T-cell markers CD2, CD3, CD5, and CD7 in patients with early-stage mycosis fungoides is associated with overall survival and progression-free survival. Methods: This retrospective included 83 patients with stage IA–IIA MF diagnosed between 2003 and 2012 at a single institution. Immunohistochemical staining of archived biopsy specimens was performed for CD2, CD3, CD5, and CD7. Loss of marker expression was defined as absence in ≥30% of lymphocytes. Clinical and histopathological data were correlated with survival and progression outcomes using Kaplan–Meier curves and log-rank tests. Results: Loss of at least one T-cell marker was identified in 66% of patients, most commonly CD7 (72%), followed by CD5 (11%) and CD2 (11%). No cases showed loss of CD3 expression. CD7 loss was significantly associated with shorter progression-free survival (*p* < 0.05), but not with overall survival. No significant associations were found between CD2 or CD5 loss and either survival or disease progression. Conclusions: CD7 loss was the only immunohistochemical abnormality significantly associated with earlier disease progression in early-stage MF, suggesting a potential prognostic role. In contrast, loss of CD2 and CD5 did not affect survival or progression, and CD3 was preserved in all cases. These findings highlight the value of incorporating CD7 status into prognostic assessment, although larger studies are needed to confirm its utility.

## 1. Introduction

Mycosis fungoides (MF) is the most common type of primary cutaneous T-cell lymphoma, accounting for approximately 50% of all cases [1,2,3,4]. It is characterized by the proliferation of malignant, skin-homing CD4+ T-cells that preferentially infiltrate the epidermis and superficial dermis. MF exhibits a chronic, often indolent course in its early phases, but can progress over time to more aggressive forms. Clinically, it evolves through three recognized stages, patch, plaque, and tumor, each of which can present with distinct histopathological features [1,4].

From an epidemiological perspective, MF primarily affects adults, typically presenting between the fifth and sixth decades of life, with a higher incidence in males. Despite its classification as a malignancy, early-stage MF tends to have a favorable prognosis, with overall survival rates comparable to that of the general population in patch- or plaque-limited disease. However, in advanced stages—particularly in the tumor stage, or cases that evolve into Sézary syndrome or transform into large cell lymphoma—the disease becomes markedly more aggressive and prognostically poor [1,3,4].

Histologically, early-stage MF is characterized by epidermotropism of small-to-medium-sized atypical T-cells with cerebriform nuclei, often arranged in clusters known as Pautrier microabscesses. However, these features may be subtle or absent, particularly in early lesions, and inflammatory mimics such as chronic dermatitis, eczema, or parapsoriasis en plaque can closely resemble MF on both clinical and histologic grounds. In plaque and tumor stages, the atypia becomes more pronounced, with denser dermal infiltrates, possible loss of epidermotropism, and, in some cases, large cell transformation [1,4].

Because of this diagnostic complexity, immunohistochemistry (IHC) plays an essential role in confirming MF and excluding mimics. Immunophenotyping typically demonstrates a predominance of CD3+, CD4+, and CD45RO+ T-cells, with loss of pan-T-cell markers such as CD5, CD7, or CD26. The absence of B-cell markers (e.g., CD20) helps to differentiate MF from B-cell lymphoproliferative disorders. Molecular techniques including T-cell receptor (TCR) gene rearrangement studies have been increasingly employed to support the diagnosis, particularly in early or ambiguous cases [1,3,4].

The EORTC has also published comprehensive treatment recommendations, including their most recent 2023 consensus guidelines for MF and Sézary syndrome [3]. These recommendations stress the importance of stage-based therapeutic strategies, highlighting skin-directed therapies such as topical corticosteroids, phototherapy (e.g., PUVA or narrowband UVB), and topical chemotherapeutics (e.g., nitrogen mustard) for early-stage disease. In contrast, more advanced stages may necessitate systemic therapies including interferon-alpha, bexarotene, histone deacetylase inhibitors, or, in selected cases, hematopoietic stem cell transplantation [3].

The 2024 EORTC update on the classification of cutaneous lymphomas further clarifies the histopathological subgroups and refines diagnostic thresholds, particularly for early MF, folliculotropic MF, pagetoid reticulosis, and granulomatous slack skin—all variants of MF with differing clinical behavior and therapeutic response profiles [4]. This refinement is critical for dermatopathologists and clinicians alike, as it enables more precise risk stratification and individualized patient management.

It typically follows a chronic, progressive course through patch, plaque, and tumor stages. Early diagnosis is often challenging, as its histological features overlap with those of inflammatory dermatoses [2,5,6,7]. Immunohistochemistry (IHC) is therefore essential for diagnosis and prognostic assessment [8,9,10,11].

MF is typically composed of epidermotropic α/β T-helper memory lymphocytes expressing CD3+, CD4+, CD5+, CD45RO+, and βF1+, while lacking CD8 expression [7,12,13,14]. In advanced stages, partial or complete loss of pan-T-cell markers (CD2, CD3, CD5) and CD4/CD8 expression is common, reflecting a neoplastic T-cell origin [3,12,13,15,16,17]. Some cases show aberrant phenotypes such as CD4−/CD8− or CD4+/CD8+ combinations, and may coexpress follicular helper T-cell markers, including PD-1, BCL6, CXCL13, and CD10 [11,18,19].

Loss of CD7, observed in <10% of T-cells in early lesions, is considered suggestive of MF, though not specific, as it may also occur in some inflammatory dermatoses [12,20,21,22]. In the tumor stage, some MF cases acquire cytotoxic markers such as TIA-1 and granzyme B, indicating a phenotypic shift without transforming into cytotoxic lymphomas [23]. Rarely, CD56 is expressed in cytotoxic-phenotype variants [12,24,25,26].

CD30 expression in plaque- and tumor-stage MF is variably associated with prognosis. CD30+ large cell transformation may suggest an indolent course, while CD30– transformations are often associated with worse outcomes, though the prognostic value of CD30 remains debated [15,27,28,29]. FOXP3+ regulatory T-cells, typically present in early-stage lesions, decrease with disease progression, and may hold prognostic significance [14,24,25,30,31,32,33,34,35]. Additional cell types, such as CD1a+ Langerhans cells and CD20+ B-cells, may appear in lesions, complicating diagnosis [12,13,36,37,38,39,40].

Although IHC alone is insufficient for definitive diagnosis, it offers crucial supportive evidence when combined with histopathological and clinical data, and enables tracking of phenotypic changes that may signal disease progression or transformation. In summary, MF remains a diagnostic and therapeutic challenge despite being the most prevalent form of CTCL. Its clinical presentation can mimic benign dermatoses, while its histopathology may overlap with non-neoplastic inflammatory conditions, especially in early stages. The continued evolution of classification systems by the EORTC and WHO—combined with advances in immunohistochemistry and molecular diagnostics—has significantly improved our ability to accurately identify and manage this disease. However, early recognition remains a priority, and greater awareness of its histological variability is essential. The importance of integrating clinicopathologic features with ancillary studies cannot be overstated, particularly given the prognostic and therapeutic implications of early versus advanced disease. As new consensus guidelines emerge and classification systems are refined, dermatopathologists are uniquely positioned to play a central role in advancing the accurate diagnosis and understanding of MF [14,29,33].

Despite advancements in immunophenotyping, no single immunohistochemical marker reliably diagnoses early-stage MF or predicts disease course. This study aimed to determine whether loss of CD2, CD3, CD5, or CD7 correlates with clinical parameters, including age, sex, premycotic phase duration, treatment response, and relapse frequency.

## 2. Methods

This retrospective, single-center study included patients with a histopathologically and clinically confirmed diagnosis of early-stage mycosis fungoides (MF), corresponding to stages IA, IB, and IIA, as defined by the TNMB classification system established by the International Society for Cutaneous Lymphomas (ISCL) in collaboration with the European Organization for Research and Treatment of Cancer (EORTC). The TNMB staging criteria were applied consistently across all cases to ensure diagnostic uniformity and to define early-stage disease based on the extent of skin involvement (T), nodal status (N), visceral involvement (M), and blood involvement (B) [2,4,41].

Eligible patients were identified from the archives of the Clinic for Dermatovenereology and the Clinical Institute for Pathology and Cytology, both part of the Clinical Hospital Center Zagreb. The study timeframe spanned a ten-year period, including cases diagnosed between January 2003 and December 2012 [2,4,41]. Inclusion criteria were as follows: (1) diagnosis of MF confirmed by both clinical presentation and histopathological features; (2) classification within early-stage MF (IA–IIA); (3) availability of formalin-fixed, paraffin-embedded (FFPE) tissue suitable for immunohistochemical (IHC) analysis; and (4) complete clinical documentation available in patient records. Cases were excluded if tissue samples were inadequate for staining or if clinical follow-up data were incomplete or unavailable. A total of 83 patients fulfilled all inclusion criteria and were included in the final analysis cohort. Only clearly diagnosed MF cases, established according to WHO/EORTC criteria and confirmed by clinicopathological correlation, were included; PCR-based TCR-γ clonality testing was performed selectively but not routinely, and was therefore not the primary focus of this study.

Comprehensive clinical data were extracted from patient medical records. Variables collected included age at diagnosis, sex, detailed clinical presentation at initial evaluation, duration of the premycotic (premalignant or nonspecific) phase prior to diagnosis, primary and adjunctive treatment modalities used, clinical response to therapy, and the occurrence of relapse or disease progression during follow-up. Disease progression was stringently defined as any of the following events: (1) clinical or histological transition from patch/plaque- to tumor-stage MF; (2) development of erythroderma; (3) histologically confirmed lymph node involvement not present at baseline; (4) evidence of visceral dissemination; or (5) disease-related mortality. These criteria were consistent with international standards and designed to distinguish between stable and progressive disease accurately.

Archived diagnostic skin biopsy specimens were retrieved for IHC analysis. Sections were cut at 4 μm thickness from FFPE blocks and subjected to standard histological processing, including deparaffinization in xylene, rehydration through graded alcohols, and antigen retrieval. Antigen retrieval was performed in a microwave oven using Target Retrieval Solution (pH 9.0; Dako, Denmark) at 95 °C for 15 min. The slides were then immunostained with a panel of monoclonal antibodies targeting T-cell surface markers: CD2 (clone AB75, Novocastra, UK), CD3 (clone F7.2.38, Dako, Denmark), CD4 (clone 4B12, Dako), CD5 (clone 4C7, Dako), CD7 (clone CBC.37, Dako), and CD8 (clone C8/144B, Dako). The immunostaining protocol utilized the labelled streptavidin-biotin (LSAB) detection system, performed on the Dako TechMate™ automated staining platform.

For each case, the intensity and distribution of membranous staining in lymphocytes were evaluated. Staining was interpreted as positive only when clear membranous expression was observed in infiltrating lymphoid cells. Internal controls were used to confirm the validity of staining runs, and lymph node tissue was employed as an external positive control for each antibody. Negative controls included tissue sections processed identically, but omitting the primary antibody.

Loss of antigen expression was defined as the absence of membranous staining in ≥30% of infiltrating lymphocytes, which in our cohort were overwhelmingly CD3+ T lymphocytes. Quantification was performed by evaluating 10 randomly selected high-power fields (400× magnification) across lesional zones, with two independent dermatopathologists blinded to clinical outcomes performing the assessments. In cases of discrepancy, a consensus was reached via joint review.

All statistical analyses were conducted using Statistica version 6.0 (StatSoft, Tulsa, OK, USA). Descriptive statistics were used to summarize baseline demographic and clinical characteristics. Associations between immunohistochemical expression patterns and categorical clinical variables (e.g., sex, clinical stage, presence of progression) were analyzed using the chi-square test. Continuous variables (e.g., age, duration of premycotic phase) were compared using the non-parametric Mann–Whitney U test. Survival analyses, including both overall survival (OS) and progression-free survival (PFS), were conducted using the Kaplan–Meier method, and statistical significance between groups was assessed using log-rank tests. A *p*-value of less than 0.05 was considered statistically significant.

## 3. Results

A total of 83 patients with early-stage MF were included in this study, classified as stage IA (32.5%), stage IB (42.2%), or stage IIA (25.3%) according to TNMB criteria. The median clinical follow-up duration was 25 months (range: 1–130 months). Disease progression occurred in 36 patients (43%) (Figure 1), with a median progression-free survival of 48 months. Ten patients (12%) died, all following disease progression (Figure 2). The five-year overall survival rate was 86%, and the median survival time following progression was 58 months (Table 1).

Loss of one or more pan-T-cell surface markers (CD2, CD3, CD5, or CD7) was shown in Table 1.

Examples of the IHC analysis are shown in Figure 3 and Figure 4.

### 3.1. Prognostic Value of T-Cell Surface Marker Loss in Relation to Survival

Survival analysis was conducted to assess the prognostic significance of immunohistochemical loss of CD2, CD3, CD5, and CD7 in early-stage MF (Table 2). CD3 loss was not observed in any patient and was excluded from analysis. Loss of CD2 was seen in nine patients, all of whom survived during follow-up (log-rank test, *p* = 0.2349). CD5 loss occurred in nine patients, with one death, but the association with survival was not statistically significant (*p* = 0.9050).

CD7 was the most frequently lost marker, present in 60 patients (72%). Although eight of these patients died, Kaplan–Meier analysis showed no significant difference in overall survival between those with and without CD7 loss (*p* = 0.3369) (Figure 5).

These findings indicate that isolated loss of CD2, CD5, or CD7 does not significantly affect overall survival in patients with early-stage MF (Table 2).

### 3.2. Prognostic Value of T-Cell Marker Loss in Relation to Progression-Free Survival

The relationship between immunohistochemical loss of CD2, CD3, CD5, and CD7 and progression-free survival was assessed using the log-rank test (Table 3). Loss of CD2 and CD5 was not significantly associated with disease progression. Although progression occurred less frequently among patients with CD2 loss, the difference was not statistically significant (*p* = 0.1218). CD5 loss showed no meaningful association with progression (*p* = 0.9837). CD3 loss was not observed in any patient and was excluded from the analysis.

In contrast, CD7 loss was significantly associated with earlier disease progression (*p* < 0.05) (Figure 6). Patients with CD7 loss progressed more frequently and more rapidly than those with preserved CD7 expression. Among the markers evaluated, only CD7 loss emerged as a statistically significant adverse prognostic factor for progression-free survival in early-stage MF.

## 4. Discussion

This study explored the prognostic significance of immunohistochemical loss of T-cell antigens—specifically CD2, CD3, CD5, and CD7—in patients diagnosed with early-stage mycosis fungoides (MF). Aberrant antigen expression patterns are a well-recognized feature of cutaneous T-cell lymphomas (CTCLs) and are frequently used to support diagnosis, particularly in histologically ambiguous cases. Among these, CD7 loss is considered one of the earliest and most consistent immunophenotypic abnormalities seen in MF. However, while the diagnostic implications of antigen loss have been widely described, its prognostic relevance—especially in early-stage MF—remains insufficiently defined. Our study aimed to clarify whether loss of these markers correlates with clinical outcomes, including overall survival (OS) and progression-free survival (PFS), in early-stage disease. It should be noted that, while the EORTC criteria define CD7 and CD5 loss using higher cutoffs (≥90% and ≥50%, respectively), we applied a ≥30% threshold in line with earlier immunohistochemical studies, recognizing that cutoffs vary across the literature (30–90%) and that our findings reflect prognostic rather than diagnostic implications.

Among the T-cell markers examined, CD7 was the most frequently lost antigen, observed in 72% of patients within our cohort. This is consistent with prior studies that report CD7 as the earliest and most commonly downregulated marker in MF [12,42,43,44,45]. The frequent loss of CD7 has been attributed to its relatively low expression levels in normal memory T-cells, as well as to epigenetic modifications and clonal expansion of neoplastic T-cell populations. CD7 is involved in T-cell activation and adhesion, and its downregulation may reflect early steps in immune evasion and malignant transformation. Despite its high prevalence in our cohort, CD7 loss did not show a statistically significant association with overall survival, reinforcing earlier observations that antigen aberrancy in isolation—particularly in early-stage disease—does not necessarily predict mortality [42].

Interestingly, however, a statistically significant association was found between CD7 loss and reduced progression-free survival. This suggests that while CD7 aberrancy may not influence overall survival directly, it could serve as a marker of increased biological activity and disease aggressiveness. Several previous investigations have suggested that CD7-negative neoplastic T-cell populations may exhibit higher proliferative capacity or enhanced resistance to immune surveillance, potentially leading to earlier disease [44,45].

In clinical practice, the identification of CD7 loss—especially when observed diffusely and in the absence of other pan-T-cell markers—may thus warrant closer clinical monitoring and could influence therapeutic decision-making.

In contrast, loss of CD2 and CD5 did not show statistically significant associations with either OS or PFS. CD2 is a surface adhesion molecule involved in T-cell activation, and while its loss has been described in various peripheral T-cell lymphomas, its absence is less frequently observed in early-stage MF. Similarly, CD5 is a pan-T-cell marker whose downregulation has been reported more commonly in transformed or advanced MF. The lack of prognostic value for these markers in our early-stage cohort suggests that their loss may represent a later event in the disease course or may be more relevant in aggressive CTCL subtypes. These findings are in line with the previous literature, which has not consistently demonstrated prognostic implications for CD2 or CD5 loss in early MF [43]. Of particular note, CD3 expression was retained in all cases included in our study. CD3 is a highly conserved component of the T-cell receptor complex, and complete loss is considered an uncommon event in CTCL, usually associated with late-stage disease or large cell transformation. Its uniform expression across our cohort is likely reflective of the early clinical stage of the included patients. This supports the widely accepted notion that pan-T-cell antigen loss, particularly involving CD3, tends to occur predominantly in advanced or transformed MF [14,43].

The prognostic implications of immunohistochemical marker loss in MF remain an evolving area of study. While immunophenotypic aberrations are well-established diagnostic tools, their utility in guiding risk stratification and therapeutic planning is less defined. Our findings contribute to this field by identifying CD7 loss as a potential early indicator of increased disease activity, despite its lack of correlation with mortality. This distinction is clinically important, as patients with early MF often have prolonged survival, but a subset may experience rapid progression. Identifying early predictors of such progression could aid in tailoring follow-up schedules and initiating early interventions.

Limitations of our study include the retrospective nature of data collection and the relatively small sample size, which may limit the generalizability of our findings. Moreover, while immunohistochemistry provides valuable information about antigen expression, it does not fully capture the molecular complexity of CTCL. Future studies may benefit from integrating flow cytometry, T-cell receptor clonality testing, and genomic profiling to provide a more comprehensive prognostic framework. A limitation of our study is the absence of systematic clonality testing, as TCR-γ PCR was performed only in selected cases and not routinely across the cohort, and therefore could not be incorporated into the multivariate analysis.

In conclusion, our results demonstrate that, while loss of CD2, CD3, and CD5 does not confer significant prognostic value in early-stage MF, CD7 loss is significantly associated with reduced progression-free survival. This suggests that CD7 may serve not only as a diagnostic marker but also as a potential early prognostic indicator. These findings warrant further investigation in larger prospective cohorts and highlight the need for continued refinement of prognostic models in early MF, with the ultimate goal of improving patient stratification and individualized treatment planning.

## Figures and Tables

**Figure 1 dermatopathology-12-00029-f001:**
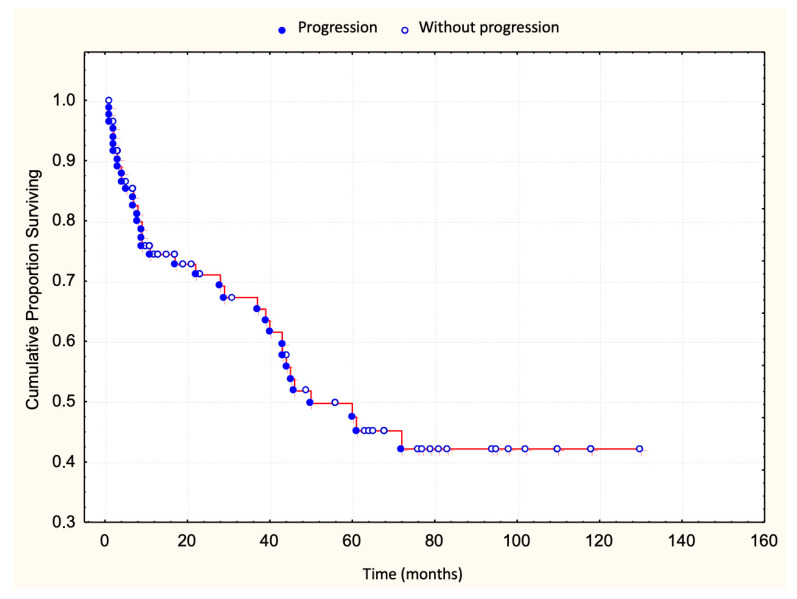
Survival curve until progression.

**Figure 2 dermatopathology-12-00029-f002:**
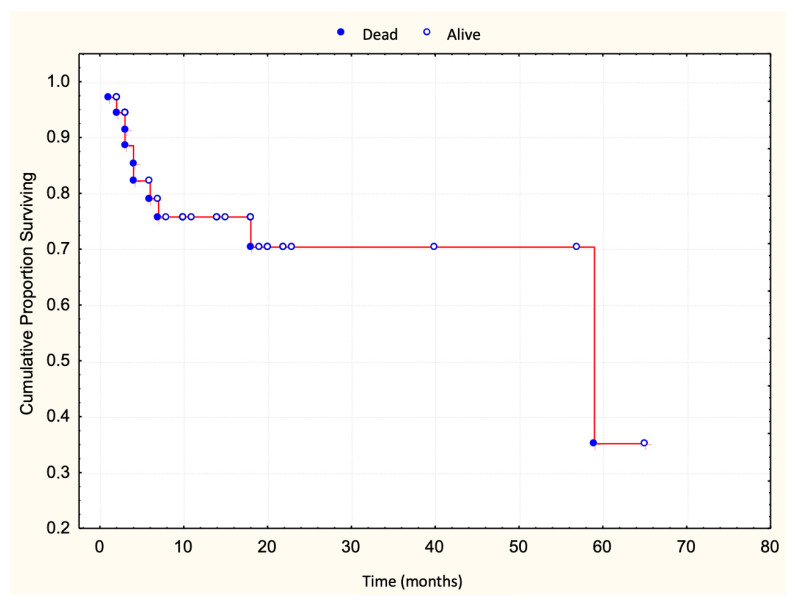
Survival curve after progression.

**Figure 3 dermatopathology-12-00029-f003:**
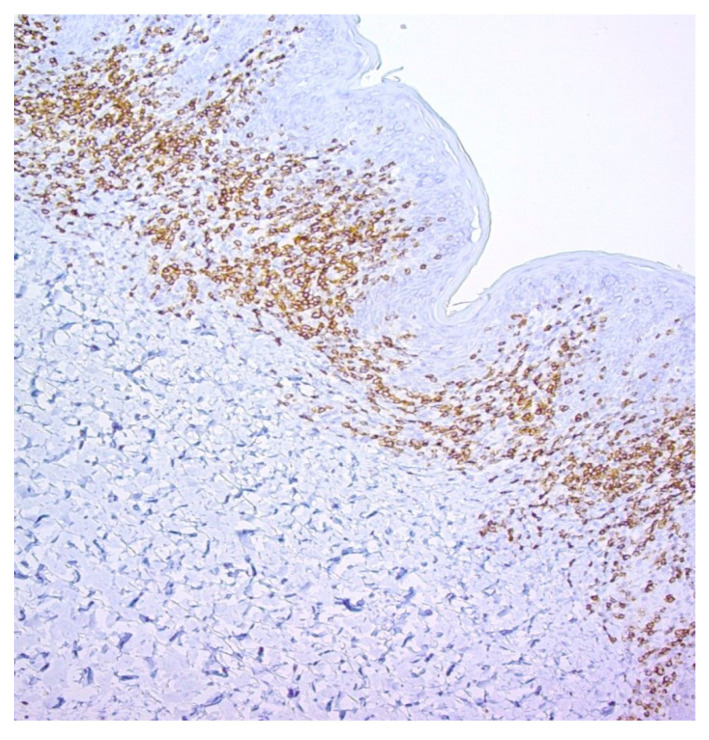
CD4 expression (positivity) in mycosis fungoides (MF). Magnification 10×. Basilar, medium-sized lymphocytes with atypical morphology showing CD4 positivity.

**Figure 4 dermatopathology-12-00029-f004:**
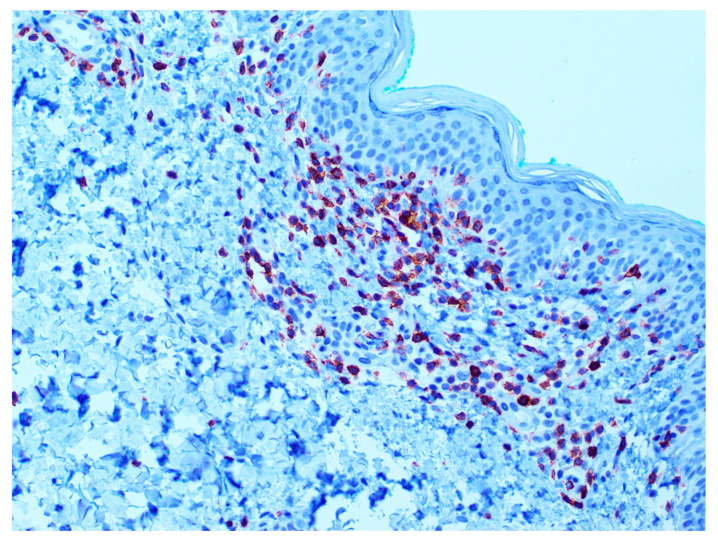
CD7 expression (positivity) in mycosis fungoides (MF). Magnification 10×. Basilar, medium-sized lymphocytes with atypical morphology showing CD7 positivity.

**Figure 5 dermatopathology-12-00029-f005:**
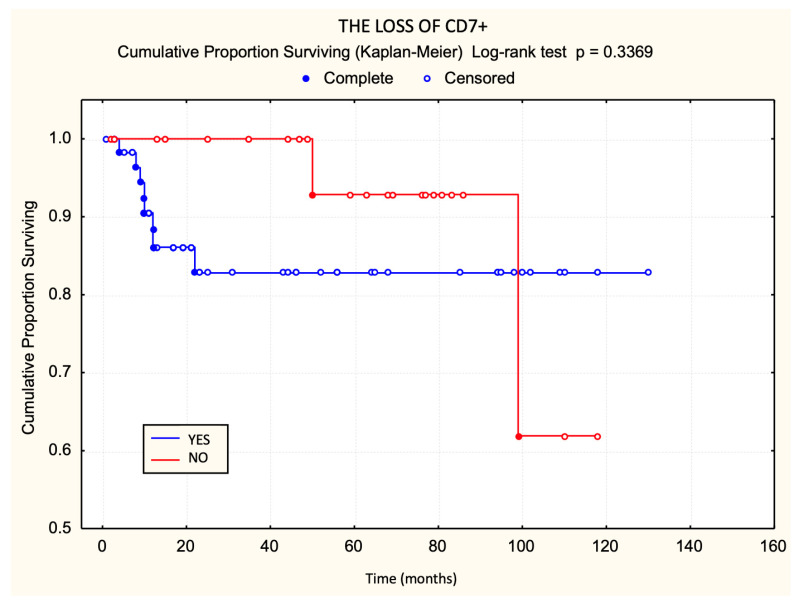
Prognostic value of CD7 loss in relation to survival.

**Figure 6 dermatopathology-12-00029-f006:**
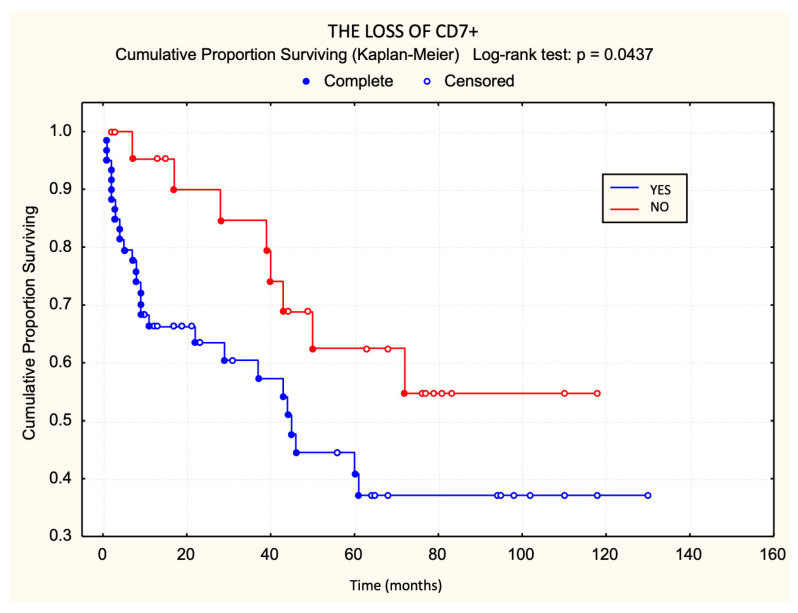
CD7 loss in relation to progression-free survival.

**Table 1 dermatopathology-12-00029-t001:** Clinical characteristics and immunohistochemical findings in patients with early-stage mycosis fungoides.

Variable	Patients (N = 83)
Age (years)	60 (7–85)
Gender	
Male	49 (59%)
Female	34 (41%)
Disease Stage	
Ia	27 (33%)
Ib	35 (42%)
IIa	21 (25%)
Disease Progression	36 (43%)
Fatal Outcome	10 (12%)
Loss of CD2	9 (11%)
Loss of CD3	0
Loss of CD5	9 (11%)
Loss of CD7	60 (72%)

**Table 2 dermatopathology-12-00029-t002:** Survival status in relation to immunohistochemical loss of T-cell markers CD2, CD3, CD5, and CD7 in patients with early-stage mycosis fungoides.

Marker	Survived N = 73	Died N = 10	Log-Rank Test *p*-Value
CD2 loss	9	0	*p* = 0.2349
CD3 loss	0	0	
CD5 loss	8	1	*p* = 0.9050
CD7 loss	52	8	*p* = 0.3369

**Table 3 dermatopathology-12-00029-t003:** Progression-free survival status in relation to immunohistochemical loss of T-cell markers CD2, CD3, CD5, and CD7 in patients with early-stage mycosis fungoides.

	No Progression	Progression	Log-Rank Test
Loss of CD2	9	0	*p* = 0.2349
Loss of CD3	0	0	-
Loss of CD5	8	1	*p* = 0.9050
Loss of CD7	52	8	*p* = 0.3369

## Data Availability

The data presented in this study are available upon reasonable request from the corresponding author. Due to privacy and ethical considerations, the data are not publicly available.

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
