# Peer review of "Prognostic Value of Immunohistochemical T-Cell Marker Loss in Early-Stage Mycosis Fungoides: A Single-Center Cohort Study"

_dermatopathology, 2025, doi:10.3390/dermatopathology12030029_

Round 1

Reviewer 1 Report

Comments and Suggestions for Authors

The authors conducted a retrospective analysis of patients with early mycosis fungoides (MF) to evaluate the prognostic implication of loss of T-cell markers and found that absence of CD7 but not CD2 and CD5 was associated with decreased progression-free survival. This is an interesting study, but I have some concerns.

  1. Introduction is too long. The general explanation of MF, the diagnostic challenge in MF and the classification of CTCL can be shortened or removed.

  1. What is “CD4 u2013/CD8 u2013” in Introduction?

  1. Why did authors define loss of antigen expression as the absence of staining in ≥30% of lymphocytes? The rationale is unclear and should be clarified. In addition, I think lymphocytes should be CD3-positive T-cells.

  1. In Figure4, the example of cases with absent CD7 expression should also be presented.

  1. The presence of plaques, N1 in TNMB classification (stage IIA), and elevated LDH levels are known to be associated with disease progression in early MF. Please check their impact on the progression in authors’ cohort. If they are associated with progression, the multivariate analysis including them and loss of CD7 should be conducted.

  1. The authors described “These findings are in line with previous literature, which has not consistently demonstrated prognostic implications for CD2 or CD5 loss in early MF” in Discussion, but no references are cited.

Author Response

Answers to comments ROUND 1

Reviewer 1: Response Letter

The authors conducted a retrospective analysis of patients with early mycosis fungoides (MF) to evaluate the prognostic implication of loss of T-cell markers and found that absence of CD7 but not CD2 and CD5 was associated with decreased progression-free survival. This is an interesting study, but I have some concerns.

  1. Introduction is too long. The general explanation of MF, the diagnostic challenge in MF and the classification of CTCL can be shortened or removed.

Response:
We thank the reviewer for this helpful comment. We have substantially shortened the Introduction by removing redundant details on MF classification and treatment guidelines, and by focusing only on background directly relevant to our study aim (the prognostic role of immunohistochemical marker loss).

 Manuscript change: Some paragraphs of the introduction have been removed.

  1. What is “CD4 u2013/CD8 u2013” in Introduction?

Response:
We apologize for the typographical error. The symbol “u2013” was a formatting error and has been corrected to “CD4⁻/CD8⁻” in the revised manuscript.

Manuscript change: Corrected in Introduction, line 139.

  1. Why did authors define loss of antigen expression as the absence of staining in ≥30% of lymphocytes? The rationale is unclear and should be clarified. In addition, I think lymphocytes should be CD3-positive T-cells.

Response:
We thank the reviewer for this comment. We agree that the most rigorous approach would be to define antigen loss relative to CD3+ T lymphocytes. In our cohort, however, the lesional infiltrates were overwhelmingly composed of CD3+ cells, and CD3 expression was preserved in all cases. Thus, evaluating loss relative to total lymphocytes and relative to CD3+ T cells would yield identical results. We have clarified this point in the Methods section to avoid ambiguity.

Manuscript change: In Methods → change sentence to:
Loss of antigen expression was defined as the absence of membranous staining in ≥30% of infiltrating lymphocytes, which in our cohort were overwhelmingly CD3+ T lymphocytes.”

Regarding the ≥30% threshold, we adopted this cutoff because it has precedent in the immunohistochemical evaluation of T-cell antigen expression in mycosis fungoides. Specifically, Michie et al. (1990, Am J Pathol, 137:1447–1451, PMCID: PMC1877727) used ≥30% as the cut-off for discordant antigen expression, based on reproducibility studies of lymphocyte subset estimation in tissue sections. Although other studies have used alternative thresholds (e.g., 50%), we chose 30% to align with this established methodology and to avoid overinterpretation of focal or artifactual negativity.

  1. In Figure4, the example of cases with absent CD7 expression should also be presented.

We thank the reviewer for this suggestion. Our intention in Figure 4 was to provide a visual example of preserved CD7 expression to help orient readers to the staining pattern. The cases with loss of CD7 expression are described in detail in the Results and shown in the survival analyses. To clarify this, we have revised the figure legend to explicitly state that Figure 4 illustrates CD7 positivity, while representative cases with absent CD7 expression are described in the text.

  1. The presence of plaques, N1 in TNMB classification (stage IIA), and elevated LDH levels are known to be associated with disease progression in early MF. Please check their impact on the progression in authors’ cohort. If they are associated with progression, the multivariate analysis including them and loss of CD7 should be conducted.

We thank the reviewer for this important suggestion. We would like to note that the clinical prognostic associations of plaque stage were comprehensively analyzed and published as part of this project in a separate manuscript: Dermatopathology 2024, 11(2), 161–176. https://doi.org/10.3390/dermatopathology11020017. LDH levels were not investigated in this study due to its retrospective nature.

  1. The authors described “These findings are in line with previous literature, which has not consistently demonstrated prognostic implications for CD2 or CD5 loss in early MF” in Discussion, but no references are cited.  

We thank the reviewer for this observation. We have now added supporting references into discussion (line 3385386, marked yellow)

Reviewer 2 Report

Comments and Suggestions for Authors

The paper is well written and seeks to answer an interesting question of whether phenotypic aberrancy is associated with overall survival and PFS, but it is not acceptable in the current form.  

I have some further questions and concerns that I would like to have clarified before considering the paper for publication.

A major point that is made by the authors in this manuscript involves the significance of loss of CD7 expression, and using the criteria set for by the authors loss of staining by tumor cells is associated with reduced PFS but not OS.  The authors use a cutoff of "absence of staining in > or = 30%".  Whereas, the EORTC criteria for phenotypic aberrancy/loss of CD7 to be absence of staining in > or = 90%.  This is a significant difference.  The threshold was set very high because absence of CD7 staining may be seen in malignant and benign inflammatory disorders, and the lower threshold will increase the number of cases included, with reduced specificity for mycosis fungoides.  Furthermore , CD7 is often a low intensity stain, making it often difficult to quantify with exactness to the nearest 10th percentiles.  Similarly, loss of CD5 expression was defined as > or = 30%, whereas the commonly accepted criteria for this marker is 50%.   I request that the authors clarify in the Methods section, how these cutoffs were obtained, and the rationale for their use.  I question the reproducibility of distinguishing loss of CD7 expression in 20% (not aberrant in this study) vs 30% (aberrant by definition in this study) given the challenges in interpretation as stated above.   

Additional recommendations:

Line 97:  Would add CD5 to CD7 or CD26 as it is used in this study and is commonly used in clinical practice.

Line 99:  Next gen sequencing is not often used on early stage MF lesions given the low tumor density in the biopsy and high expected rate of failure of the assay due to insufficient template.  I would either clarify how Next gen is used in this context or modify the statement to include other assays instead such as high throughput TCR sequencing with greater analytical sensitivity. 

Line 128:  Although most cases of MF are correctly stated to be CD4+, CD8+ variants are not rare, and this immunophenotype has been shown not to influence the prognosis.  

Line 144:  Increased intratumoral CD8+ cells being linked to a greater risk of progression and reduced PFS is a controversial one.  Although it was correctly stated by the cited reference, a prior paper by Hoppe, Medeiros et al (JAAD 1995) in a larger cohort of 78 patients and with stage adjusted analysis (the cited paper seemed to have a bias toward plaque/tumor over patch disease) demonstrated an improved prognosis with higher CD8+ intratumoral lymphocytes.   I recommend this be acknowledged as a debatable subject.  

Author Response

Reviewer 2 – Response Letter

Comment 1 (Major):
A major point that is made by the authors in this manuscript involves the significance of loss of CD7 expression, and using the criteria set for by the authors loss of staining by tumor cells is associated with reduced PFS but not OS. The authors use a cutoff of "absence of staining in > or = 30%". Whereas, the EORTC criteria for phenotypic aberrancy/loss of CD7 to be absence of staining in > or = 90%. This is a significant difference. The threshold was set very high because absence of CD7 staining may be seen in malignant and benign inflammatory disorders, and the lower threshold will increase the number of cases included, with reduced specificity for mycosis fungoides. Furthermore , CD7 is often a low intensity stain, making it often difficult to quantify with exactness to the nearest 10th percentiles. Similarly, loss of CD5 expression was defined as > or = 30%, whereas the commonly accepted criteria for this marker is 50%. I request that the authors clarify in the Methods section, how these cutoffs were obtained, and the rationale for their use. I question the reproducibility of distinguishing loss of CD7 expression in 20% (not aberrant in this study) vs 30% (aberrant by definition in this study) given the challenges in interpretation as stated above.

Response:
We thank the reviewer for this important observation, which was also raised by Reviewer 1. We have now substantially clarified this issue in the Methods section. We agree that the EORTC diagnostic criteria define phenotypic aberrancy of CD7 as absence of staining in ≥90% of tumor cells, and CD5 as ≥50%. However, our aim was not to apply the EORTC cutoff for diagnostic purposes but rather to investigate prognostic implications in a cohort of confirmed MF cases. For this purpose, we used the ≥30% cutoff, which has precedent in earlier immunohistochemical studies (Michie et al., Am J Pathol 1990, doi:10.1016/S0002-9440(10)65093-6; Cotta et al., São Paulo Med J 2004, doi:10.1590/S1516-31802004000400006).

We also recognize the practical challenges of quantifying CD7 staining with precision, as the reviewer points out. To address this, we have explicitly acknowledged in Discussion that reproducibility may be limited around the threshold, that cutoffs vary across the literature (30–90%), and that our findings relate specifically to prognosis rather than diagnosis.

We have inserted this paragraph into discussion:”It should be noted that while the EORTC criteria define CD7 and CD5 loss using higher cutoffs (≥90% and ≥50%, respectively), we applied a ≥30% threshold in line with earlier immunohistochemical studies, recognizing that cutoffs vary across the literature (30–90%) and that our findings reflect prognostic rather than diagnostic implications.” Line 356-359

Comment 2 (Line 97):
Would add CD5 to CD7 or CD26 as it is used in this study and is commonly used in clinical practice.

Response:
We agree and have revised the text at line 97 to include CD5 alongside CD7 and CD26, as CD5 is both evaluated in this study and commonly used in clinical practice. Now line 76

Comment 3 (Line 99):
Next gen sequencing is not often used on early stage MF lesions given the low tumor density in the biopsy and high expected rate of failure of the assay due to insufficient template. I would either clarify how Next gen is used in this context or modify the statement to include other assays instead such as high throughput TCR sequencing with greater analytical sensitivity.

Response:
We thank the reviewer for this comment. We agree that next-generation sequencing (NGS) is limited by low tumor burden in early MF and may have a high failure rate. We have revised the text: “Molecular techniques including T-cell receptor (TCR) gene rearrangement studies have been increasingly employed to support the diagnosis, particularly in early or ambiguous cases.”

Comment 4 (Line 128):
Although most cases of MF are correctly stated to be CD4+, CD8+ variants are not rare, and this immunophenotype has been shown not to influence the prognosis.

Response:
We thank the reviewer for this clarification. The text has been revised: “Some cases show aberrant phenotypes such as CD4-/CD8- or CD4+/CD8+ combinations, and may coexpress follicular helper T-cell markers, including PD-1, BCL6, CXCL13, and CD10.”

Comment 5 (Line 144):
Increased intratumoral CD8+ cells being linked to a greater risk of progression and reduced PFS is a controversial one. Although it was correctly stated by the cited reference, a prior paper by Hoppe, Medeiros et al (JAAD 1995) in a larger cohort of 78 patients and with stage adjusted analysis (the cited paper seemed to have a bias toward plaque/tumor over patch disease) demonstrated an improved prognosis with higher CD8+ intratumoral lymphocytes. I recommend this be acknowledged as a debatable subject.

Response:
We thank the reviewer for highlighting this important point. We acknowledge that the prognostic impact of intratumoral CD8+ cells is indeed controversial, with conflicting evidence in the literature. Since this topic was not the primary focus of our study, and in order to avoid overstating conclusions outside the scope of our work, we have removed this sentence from the Introduction.

Reviewer 3 Report

Comments and Suggestions for Authors

Interesting article. A few important points should be addressed:

  1. Please include the recent EORTC efforts to distinguish between patch and plaque in MF (DOI: 10.1111/jdv.18852). How were these lesions differentiated in your study? This information MUST be included in the patient population. 

  2. How were diagnoses established, particularly regarding the differential with benign inflammatory dermatoses? This can be especially challenging in stage IA. In addition, how was clonality assessed?

  3. A key limitation of the study is the lack of adjustment for potential competing variables. Variables such as lesion type (patch vs plaque) or clonality (monoclonal vs polyclonal infiltrate) should be taken into account. Given that 36 patients progressed (43% of the cohort), a simple multivariate Cox model would be feasible. At minimum, an adjusted model including CD7 loss, lesion type, and site/clonality should be presented. Otherwise, the observed results may simply reflect masked effects related to lesion type or site.

  4. Was folliculotropism assessed? This feature is known to carry prognostic significance (DOI: 10.2147/CCID.S273063).

Author Response

Reviewer 3 – Response Letter

Comment 1:
Please include the recent EORTC efforts to distinguish between patch and plaque in MF (DOI: 10.1111/jdv.18852). How were these lesions differentiated in your study? This information MUST be included in the patient population.

Response:
We thank the reviewer for this important point. Our manuscript cites all most important EORTC classifications and the EORTC consensus (ref. 2, 5, 4, 41).  In our cohort, lesions were classified clinically by experienced dermatologists, with plaques defined as infiltrated lesions with elevated surface compared to flat patches. This clarification has been added to the patient population description in the Methods section.Please note also that clinical prognostic associations of plaque stage were comprehensively analyzed and published as part of this project in a separate manuscript: Dermatopathology 2024, 11(2), 161–176. https://doi.org/10.3390/dermatopathology11020017. I

Inserted part in Methods: According to EORTC classification, lesions were clinically defined by experienced dermatologists, with patches characterized as flat erythematous lesions and plaques as infiltrated lesions with elevated surface compared to patches.

Comment 2:
How were diagnoses established, particularly regarding the differential with benign inflammatory dermatoses? This can be especially challenging in stage IA. In addition, how was clonality assessed?

Response:
We agree with the reviewer that distinguishing early MF from benign inflammatory dermatoses is challenging, particularly in stage IA. In our cohort, diagnoses were established by expert dermatopathologists according to WHO/EORTC criteria, integrating clinical, histopathological, and immunophenotypic findings. In difficult early cases, close clinicopathological correlation was essential. PCR-based TCR-γ gene rearrangement studies were performed selectively, but not routinely, due to the retrospective nature of the study and because this method was not part of standard diagnostic practice at the time. Importantly, we did not include borderline or indeterminate cases; only clearly diagnosed cases confirmed by clinical, histopathological, and immunohistochemical features were included. We have added these clarifications to the Methods section and acknowledged the absence of systematic clonality testing as a limitation in the Discussion. 

Lines inserted:

Methods 246-248 : “Only clearly diagnosed MF cases, established according to WHO/EORTC criteria and confirmed by clinicopathological correlation, were included; PCR-based TCR-γ clonality testing was performed selectively but not routinely and was therefore not the primary focus of this study.”

Discussion 446 -448: “A limitation of our study is the absence of systematic clonality testing, as TCR-γ PCR was performed only in selected cases and not routinely across the cohort, and therefore could not be incorporated into the multivariate analysis.”

Round 2

Reviewer 1 Report

Comments and Suggestions for Authors

I have no further comments.

Reviewer 2 Report

Comments and Suggestions for Authors

The authors have adequately addressed the listed concerns.  I am concerned about drawing conclusions the relevance of CD7 loss based on relatively small sample sizes, which were appropriately mentioned by the authors in the discussion.